# Activated Alpha 2-Macroglobulin Is a Novel Mediator of Mesangial Cell Profibrotic Signaling in Diabetic Kidney Disease

**DOI:** 10.3390/biomedicines9091112

**Published:** 2021-08-30

**Authors:** Jackie Trink, Renzhong Li, Yaseelan Palarasah, Stéphan Troyanov, Thomas E. Andersen, Johannes J. Sidelmann, Mark D. Inman, Salvatore V. Pizzo, Bo Gao, Joan C. Krepinsky

**Affiliations:** 1Division of Nephrology, McMaster University, Hamilton, ON L8N 4A6, Canada; trinkj1@mcmaster.ca (J.T.); lirenz@mcmaster.ca (R.L.); gaolinbo@hotmail.com (B.G.); 2Unit for Thrombosis Research, Department of Regional Health Research, University of Southern Denmark, DK-6705 Esbjerg, Denmark; ypalarasah@health.sdu.dk (Y.P.); johannes.sidelmann@rsyd.dk (J.J.S.); 3Department of Medicine, Hôpital du Sacré-Coeur de Montréal, Faculty of Medicine, Université de Montréal, Montreal, QC H4J 1C5, Canada; stephan.troyanov@umontreal.ca; 4Department of Clinical Microbiology, University of Southern Denmark and Odense University Hospital, DK-5230 Odense, Denmark; thandersen@health.sdu.dk; 5Firestone Institute for Respiratory Health, Department of Medicine, McMaster University, Hamilton, ON L8N 1Y3, Canada; inmanma@mcmaster.ca; 6Department of Pathology, Duke University Medical Center, Durham, NC 27710, USA; salvatore.pizzo@duke.edu

**Keywords:** alpha 2-macroglobulin, cell signaling, cell surface GRP78, diabetic kidney disease, fibrosis, mesangial cell, PI3K/Akt signaling

## Abstract

Diabetic kidney disease (DKD) is caused by the overproduction of extracellular matrix proteins (ECM) by glomerular mesangial cells (MCs). We previously showed that high glucose (HG) induces cell surface translocation of GRP78 (csGRP78), mediating PI3K/Akt activation and downstream ECM production. Activated alpha 2-macroglobulin (α2M*) is a ligand known to initiate this signaling cascade. Importantly, increased α2M was observed in diabetic patients’ serum, saliva, and glomeruli. Primary MCs were used to assess HG responses. The role of α2M* was assessed using siRNA, a neutralizing antibody and inhibitory peptide. Kidneys from type 1 diabetic *Akita* and *CD1* mice and human DKD patients were stained for α2M/α2M*. α2M transcript and protein were significantly increased with HG in vitro and in vivo in diabetic kidneys. A similar increase in α2M* was seen in media and kidneys, where it localized to the mesangium. No appreciable α2M* was seen in normal kidneys. Knockdown or neutralization of α2M/α2M* inhibited HG-induced profibrotic signaling (Akt activation) and matrix/cytokine upregulation (collagen IV, fibronectin, CTGF, and TGFβ1). In patients with established DKD, urinary α2M* and TGFβ1 levels were correlated. These data reveal an important role for α2M* in the pathogenesis of DKD and support further investigation as a potential novel therapeutic target.

## 1. Introduction

Diabetic kidney disease (DKD) is the leading cause of kidney failure in developed nations, with patients suffering the highest morbidity and mortality rates of any kidney failure patient group. Currently, treatment can only delay DKD progression [1,2]. Thus, there is a major need to identify new therapeutic targets. The earliest pathologic hallmarks of DKD include glomerular hypertrophy and basement membrane thickening, followed by glomerular sclerosis due to the deposition of extracellular matrix (ECM) proteins [3,4]. Glomerular mesangial cells (MCs) play a central role in the pathogenesis of DKD. While we and others have gained much insight into the molecular mechanisms involved in MC matrix synthesis in response to high glucose (HG) [5], the identification of clinically translatable targets is still much needed.

The endoplasmic reticulum (ER) chaperone 78 kDa glucose-regulated protein (GRP78) maintains proper protein folding and homeostasis within the cell. It is now recognized that in non-homeostatic conditions, such as ER stress, GRP78 can also translocate to the cell surface to act as a receptor for intracellular signaling [6]. While best studied in tumor cells, we recently showed that csGRP78 is increased by HG in MCs and in diabetic kidneys and showed its importance in mediating HG-induced profibrotic responses through PI3K/Akt signaling [7]. How HG initiates intracellular signaling through csGRP78, however, has yet to be elucidated.

α2-macroglobulin (α2M) is an abundant serum protein and panproteinase inhibitor. At 720 kDa, it is comprised of four identical 180 kDa subunits [8], each containing a bait region that is cleaved once bound by a proteinase. Upon cleavage of all subunits, a conformational change occurs, entrapping the proteinase. The resulting complex is considered the activated form of α2M, designated α2M*, in which receptor recognition sites are exposed. This allows interaction with its two identified receptors, low-density lipoprotein receptor-related protein (LRP1) and csGRP78. The binding affinity for csGRP78 is significantly higher at a Kd ~100 pM compared with a Kd in the nM range for LRP1, the predominant role of which is α2M* endocytic clearance [8,9,10].

α2M* interaction with csGRP78 has thus far been implicated predominantly in the pathogenesis of various cancers [11,12,13]. α2M* binds to a region in the NH_2_-terminal domain of csGRP78 to initiate signaling pathways that promote tumor cell proliferation and survival such as ERK1/2, p38 MAPK, PI3K/Akt, and NF-κB [11,13,14,15,16]. We previously showed that HG-induced PI3K/Akt activation and downstream matrix production in MCs requires csGRP78 [7], but the ligand that activates csGRP78 has yet to be identified. Importantly, increased expression of α2M was shown in serum and saliva of diabetic patients [17,18,19], and its transcript was recently found to increase in human DKD [20]. These studies implicate α2M in diabetes and likely DKD. However, whether α2M is activated in DKD and plays a role in its pathophysiology is unknown and thus is the focus of this study.

## 2. Materials and Methods

### 2.1. Cell Culture

Primary MCs were obtained from glomeruli of male *C57BL/6* mice (Charles River, MA, USA). Briefly, after Dynabead (Thermo Fisher, Waltham, MA, USA) perfusion, kidneys were harvested and sheared, and glomeruli were isolated using a magnet. MCs were outgrown and cultured using DMEM/20% FBS (Sigma, St. Louis, MO, USA). 1LN prostate cancer cells, which express high levels of csGRP78 [13], were cultured in RPMI 1640/10% FBS (Thermo Fisher, Waltham, MA, USA). Cells were grown at 37 °C in 95% O_2_/5% CO_2_. MCs were serum-deprived at 80% confluency in medium with 1% BSA 24 h before treatment with HG (30 mM) or mannitol (24.4 mM) as an osmotic control or methylamine-activated α2M (100 pM). The peptide sequence in GRP78 to which α2M* binds (CLIGRTWNDPSVQQDIKFL (Leu^98^-Leu^115^)) was used to block α2M* binding and thus signaling through csGRP78 [11]. The scrambled peptide GTNKSQDLWIPQLRDVFI was used as a control, with both peptides used at 100 nM (GenScript, Cedarlane, Teaneck, NJ, USA).

### 2.2. Protein Extraction and Immunoblotting 

Cells were lysed as described previously [21]. Proteins were separated using SDS-PAGE followed by immunoblotting. Antibodies used: α2M (1:1000, Bioss, Woborn, MA, USA); F-α2M, which specifically detects α2M* (generated as previously described in [22]) (1:1000); pAkt S473 (1:1000, Cell Signaling, Whitby, ON, Canada); total Akt (1:1000, Cell Signaling, Whitby, ON, Canada); LRP1 (1:1000, Abcam, Cambridge, MA, USA); GRP78(C20) (1:1000, BD Biosciences, Mississauga, ON, Canada); platelet-derived growth factor-β (PDGFR-β) (1:1000, Cedarlane, Burlington, ON, Canada); collagen IV (Col IV) (1:1000, Cell Signaling, Whitby, ON, Canada); fibronectin (FN) (1:1000, Abcam, Cambridge, MA, USA); connective tissue growth factor (CTGF) (1:1000, Santa Cruz, Dallas, TX, USA); and tubulin (1:5000, Santa Cruz, Dallas, TX, USA). Media were concentrated (Amicon Ultra 4 mL Centrifugal Filter, Sigma, St. Louis, MO, USA) and run on a non-denaturing polyacrylamide gel. Membranes were probed for both inactive α2M and the conformationally changed and more rapidly migrating α2M*. Proteins in the media could not be normalized, but each experimental well was plated to the same confluency with no apparent difference in confluency observed at the time of media collection. Equal volumes of media were concentrated and run on a non-denatured gel. Nativemark unstained protein ladder (Thermo Fisher, Waltham, MA, USA) confirmed band location. 

### 2.3. qPCR 

RNA was extracted using Trizol (Invitrogen, Carlsbad, MA, USA), with 1 μg reverse transcribed using qScript Supermix Reagent (Quanta Biosciences, Gaithersburg, MD, USA). Primers for α2M were forward 5′-CCAGGACACGAAGAAGG-3′ and reverse 5′-CACTTCACGATGAGCAT-3′. Quantitative PCR was performed using the Power SYBR Green (Applied Biosystems, Waltham, MA, USA) PCR Master Mix on the Vii 7 Real-Time PCR System (Applied Biosystems, Waltham, MA, USA). Changes in mRNA expression were determined relative to 18S using the ΔΔCt method.

### 2.4. Experimental Animals and Tissue Processing 

Two type 1 diabetic models were assessed: (1) Male type 1 diabetic *C57BL/6-Ins2^Akita^/J* mice (Jackson Laboratories, Bar Harbour, ME, USA) and their wild-type controls were sacrificed at 18, 30, and 40 weeks of age (ethics approval number 18-07-30). (2) Male *CD1* mice were uninephrectomized, followed by injection with 200 µg streptozotocin and sacrifice after 12 weeks of diabetes as previously described in the original studies (ethics approval number 14-11-48) [23]. For human studies, kidney biopsy samples with a diagnosis of DKD were obtained. Normal kidney tissues surrounding resected renal cancers were used as controls. Tissue was obtained after approval by the local research ethics board (ethics approval number 2010-159). 

For immunoblotting, samples were homogenized in tissue lysis buffer containing protease inhibitors (cOmplete Mini and PhosSTOP, Sigma, St. Louis, MO, USA) in the Bead Mill Homogenizer (Bead Ruptor Elite, Omni International, Kennesaw, GA, USA) using 1.4 mm ceramic beads (Lysing Matrix D, MP Biomedicals, Fisher Scientific, Waltham, MA, USA). After clarification, protein concentration was determined using the DC Protein Assay (Bio-Rad, Mississauga, ON, Canada).

For in situ hybridization (ISH), 4 µm paraffin-embedded sections were deparaffinized, fixed (4% paraformaldehyde), and digested (proteinase K (20 mg/mL, 5 min). After pre-hybridization in hybridization buffer (ultrapure 50% formamide, 20× SSC, 10 μg/μL yeast t-RNA, 50× Denhardt’s solution) at 53 °C, 2 h, slides were incubated with a DIG-labeled α2M probe (5′AAGTAGCTTCGTGTAGTCTCT3′, Qiagen, Toronto, ON, Canada) for 2 days. Slides were washed with 2× SSC (RT) followed by washes with 2× SSC and 0.1× SSC at 53 °C. After blocking in 1× Casein, slides were incubated with AP-coupled anti-DIG antibody (Abcam, Cambridge, MA, USA) overnight at 4 °C, developed using NBT/BCIP (Vector Laboratories, Burlington, ON, Canada), washed in PBS, dehydrated, and mounted in Faramount aqueous mounting medium (DakoCytomation, Burlington, ON, Canada). Images were quantified using Image J software, and a separate quantification of glomeruli and tubules was completed (Appendix A).

For immunohistochemistry (IHC), 4 µm paraffin-embedded kidney sections were deparaffinized and then probed for α2M (Bioss, Woborn, MA, USA, 1:1000, no antigen retrieval) or α2M* (Fα2M antibody, 1:100, antigen retrieval using proteinase K, 40 µg/mL, 5 min). Images were quantified using Image J software. Separate glomerular and tubule quantification was also completed (Appendix A).

For immunofluorescence (IF), 4 µm OCT-preserved kidney sections were fixed (3.7% paraformaldehyde) and permeabilized (0.2% Triton X-100). To block endogenous biotin and reduce high background fluorescence, an Avidin/Biotin Blocking Kit (Vector Labs, Burlington, ON, Canada) was used, followed by co-staining with F-α2M (1:200) and α8-integrin as an MC marker (1:100, Novus Biologicals, Littleton, CO, USA). Images were captured using the Olympus BX41 microscope at 40×. For each mouse, 40 glomerular images were taken for quantification. The Image J colocalization plug-in was used to create a colocalization mask of areas expressing both α8-integrin and Fα2M. Quantification was completed using Image J. 

### 2.5. Transfection

For siRNA experiments, MCs were plated at 50% confluence and transfected with 100 nmol of α2M, LRP1, or control siRNA (Silencer Select, Thermo Fisher, Waltham, MA, USA) using Lipofectamine 2000 (Thermo Fisher, Waltham, MA, USA). Electroporation was used to transfect cells with pcDNA3.1 GRP78∆KDEL (GRP78 lacking the KDEL domain which localizes it to the ER, thus enabling significant localization to the cell surface [24]). Empty vector was used as a control. Confluent MCs were trypsinized and centrifuged in a medium with 20% FBS without antibiotics. Cells (200 µL, 5 × 10^5^/mL) were electroporated in a 4 mm gap cuvette with 10 µg plasmid for one 30 ms pulse at 250 V (ECM 399, BTX Harvard Apparatus, Holliston, MA, USA) before replating. MCs were then serum-deprived as above.

### 2.6. Intracellular Calcium Assay 

1LN cells were loaded with the calcium indicator Fura-2AM (5 µM, Abcam, Cambridge, MA, USA) in HBSS for 45 min at 37 °C in the dark. Baseline fluorescence readings were taken every minute for 5 min using a temperature-controlled fluorescent microplate reader (Gemini EM Spectra Max, Molecular Devices, San Jose, CA, USA) set to 340 and 380 nm excitation and 510 nm emission. After treatment with methylamine-activated α2M (100 pM) with or without antibody (Fα2M or control IgG, 2 µg), peptide or scrambled peptide (100 nM), readings were taken every minute for 15 min. Intracellular calcium concentrations were determined by calculating the ratio of the fluorescence signal (340/380 nm).

### 2.7. Cell Surface Protein Isolation 

After treatment, cells were incubated with 1 mg/mL EZ-link Sulfo-Biotin (Thermo Fisher, Waltham, MA, USA) for 30 min, then washed with 0.1 M glycine in PBS to remove excess Sulfo-Biotin, lysed, and clarified, and equal quantities of protein were incubated overnight in a 50% Neutravidin slurry (ThermoFisher, Waltham, MA, USA) to capture biotin-tagged proteins. Beads were washed 5× with lysis buffer and bound proteins cleaved by boiling for 10 min in 2× PSB. Samples were separated using SDS-PAGE and immunoblotted.

### 2.8. TGFβ1 ELISA

Total secreted TGFβ1 in conditioned media was quantified using the TGFβ1 Quantikine ELISA Kit (R&D Systems, Minneapolis, MN, USA).

### 2.9. Recovery of α2M* from Urine Samples

Urine α2M* was detected by ELISA using the Fα2M antibody as described previously [22]. This assay has been tested on plasma/serum samples. We tested the recoverability of α2M* in urine spiked with 0, 1, or 3 µg/mL α2M*. Samples were diluted to 1:100, and recovery was calculated as a percentage of the mean values. Spiking urine with 1 µg/mL α2M* revealed a recovery percentage of 109%, and the addition of 3 µg/mL showed a recovery percentage of 104%. Parallelism between the standard used in the assay and the urine plasma pool spiked with 50 µg/mL α2M* was then studied. Samples were diluted to 1:100 in sample dilution buffer and further 2-fold diluted on the plate. Values are shown in the results section.

### 2.10. Patient Cohort 

We studied the relationship between urinary α2M* and TGFβ1 using samples from a published cohort of type 2 diabetic patients with overt DKD who previously participated in a longitudinal biomarker study [25]. First, we explored whether urinary α2M* was associated with total proteinuria in a sample from 4 subjects with proteinuria <0.5 g/g creatinine and 4 with >2 g/g. Second, we verified the association between urinary α2M* and TGFβ1 in 18 subjects with proteinuria <2 g/g to attenuate the influence of proteinuria, which is also known to correlate with urinary TGFβ1. Patients had provided multiple urinary specimens during their follow-up, and α2M* was determined for each available sample. Urine TGFβ1 was previously assessed by ELISA (Millipore, Burlington, MA, USA) [25]. Urine α2M* was detected by ELISA as described above. Values were normalized to urine creatinine.

### 2.11. Statistical Analysis

A student’s *t*-test or one-way ANOVA was used to compare the means between two or more groups, respectively. For calcium assay quantification, fold change at time of treatment (6 min) was compared between groups using a one-way ANOVA. Significant differences between multiple groups (*post hoc*) were analyzed using Tukey’s HSD, with *p* ≤ 0.05 considered significant. Data are presented as mean ± SEM. To assess the relationship between urinary biomarkers in patients with DKD, we used Spearman’s Rho or Pearson correlations, depending on whether the data had skewed or normal distributions.

## 3. Results

### 3.1. α2M Is Increased and Activated by HG in MCs and in Diabetic Kidneys

Previous reports showed that α2M is increased in the saliva and serum of diabetic patients, and transcript levels are elevated in diabetic kidneys [17,18,19]. We first determined whether HG increased α2M transcript and protein expression in MCs. Figure 1A,B show that HG, but not the osmotic control mannitol, increased α2M mRNA and protein expression, respectively, and that this increase was dose-dependent (Figure 1C). We next confirmed increased α2M expression in type 1 diabetic *Akita* kidneys by ISH (Figure 1D). Appendix A shows separate quantification of glomerular and tubular staining. As shown in Figure 1E, increased α2M protein levels were also seen by IHC in glomeruli as well as in tubules (with Appendix A showing separate quantification of glomerular and tubular staining). This was confirmed by immunoblotting of kidney lysates (Figure 1F).

We next determined whether α2M is also activated in HG, thereby revealing the binding site for csGRP78. Since activation entails conformational change, we used a non-denaturing gel to preserve α2M* tertiary structure. α2M* was detected with F-α2M, an antibody that specifically recognizes the conformationally-revealed receptor-binding domain in α2M* [22]. Figure 2A shows that both secreted α2M and α2M* were significantly increased by HG, with α2M* unaffected by mannitol. As expected, α2M* migrated further through the gel. We next determined whether α2M* was increased in diabetic kidneys. As previously observed for α2M, α2M* was also elevated in both glomeruli and tubules in *Akita* diabetic kidneys, as shown by IHC (Figure 2B; Appendix A shows separated glomerular and tubular quantification). This was supported by a second model of type 1 DKD (streptozotocin-treated uninephrectomized *CD1* mice) (Figure 2C; Appendix A shows separated glomerular and tubular quantification). To confirm localization to MCs, we performed IF with dual staining for α2M* and the MC marker α8-integrin [26] in *Akita* kidneys. Using a colocalization mask, an increase in α2M* was seen in the mesangium (Figure 2D). Finally, to determine whether α2M* is increased in human DKD, its expression was assessed by IHC in kidney biopsies with a DKD diagnosis in comparison to normal kidney tissue taken at the time of renal cancer resection. Figure 2E shows that α2M*, not seen in control kidneys, was markedly expressed in diabetic glomeruli (Appendix A shows the separated quantification for glomeruli and tubules). Taken together, these data confirm that α2M expression and activation are induced by HG.

### 3.2. Inhibition of α2M/α2M* Inhibits HG-Induced Profibrotic Responses by MCs 

Previous studies identified the importance of PI3K/Akt signaling in HG-induced ECM protein production by MCs. As we previously showed that csGRP78 mediates this signaling pathway [7,27], we wished to determine whether α2M* could be the ligand leading to its activation. We thus first investigated the effects of α2M downregulation using siRNA on HG-induced PI3K/Akt activation. As seen in Figure 3A, α2M knockdown prevented activation of Akt, assessed by its phosphorylation on S473, in response to HG. Knockdown also significantly reduced ECM protein expression (fibronectin and collagen IV) and that of the profibrotic cytokine CTGF, known to contribute to mesangial expansion and kidney fibrosis in DKD [28] (Figure 3B).

We then evaluated the importance of α2M* using the F-α2M antibody. As it binds the α2M*-specific receptor binding domain, through which it binds to csGRP78, it was used here to functionally neutralize α2M* [22]. To confirm its neutralizing ability, we used 1LN prostate cancer cells, which highly express csGRP78 and in which α2M* was shown to induce a rapid increase in intracellular calcium [13]. Figure 3C shows that F-α2M, but not control IgG, prevented calcium internalization, confirming its neutralizing ability. We then tested its effects in HG. As seen in Figure 3D,E, α2M* neutralization inhibited Akt activation and ECM/CTGF upregulation, similar to α2M knockdown.

Previous studies showed that the receptor-binding domain of α2M* binds to the sequence Leu^98^-Leu^115^ in GRP78 [11]. Calcium signaling in 1LN cells induced by α2M* was abolished by a peptide comprising these residues [11,12,29]. Figure 4A confirms that this peptide, but not a scrambled control peptide, inhibits the α2M*-induced increase of intracellular calcium in 1LN cells. We thus used this peptide to confirm that α2M* mediates the HG-induced profibrotic response in MCs. As seen in Figure 4B,C, this peptide abrogated HG-induced Akt activation and matrix protein/CTGF upregulation. Scrambled peptide had no effect. Taken together, these data support the importance of α2M and its activation in mediating profibrotic responses to HG, likely through csGRP78.

### 3.3. LRP1 Is Not Involved in HG-Induced Profibrotic Responses in MCs 

As described above, both LRP1 and csGRP78 are receptors for α2M*, although its affinity is significantly higher for csGRP78 [11]. To determine whether LRP1 is involved in mediating the HG response, the effect of LRP1 knockdown using siRNA was evaluated. As seen in Figure 5A,B, this did not affect HG-induced Akt activation or matrix/CTGF upregulation, indicating that csGRP78 rather than LRP1 mediates α2M* profibrotic signals in HG.

### 3.4. Increased Matrix Synthesis with csGRP78 Overexpression Requires α2M*

In prostate cancer cells, α2M* increased csGRP78 [30]. We tested whether this positive feedback loop also occurs in MCs using a biotinylation assay to detect csGRP78. As shown in Figure 6A, α2M* increased csGRP78 in the absence of HG. We further assessed whether HG-induced Akt activation could be augmented by α2M* co-treatment (Figure 6B). However, no additive effect was seen, suggesting that the HG-induced increase in α2M* is sufficient to generate enough ligand to occupy available csGRP78. 

We next wished to evaluate whether forced GRP78 surface translocation can initiate signaling or augment HG responses. We thus overexpressed GRP78 lacking the ER-retention signal KDEL (GRP78△KDEL). We first confirmed by biotinylation and pull-down of cell surface proteins that its overexpression increased csGRP78 (Figure 6C). Interestingly, a further increase was seen with HG. We next examined its effects on Akt activation. As seen in Figure 6D, overexpression itself led to Akt activation, and this was further augmented by the addition of HG (Figure 6D). Similar effects were seen for ECM protein and CTGF expression (Figure 6E). These data suggest that in the absence of HG, basal levels of α2M* (as seen in Figure 2A) induce signaling through overexpressed csGRP78. To test this, we used the α2M* blocking peptide. As shown in Figure 6F, this blocked the Akt activation and matrix synthesis induced by csGRP78 overexpression. Similar results were seen with α2M* neutralization using the Fα2M antibody (Figure 6G). Taken together, these data support a critical role for α2M* in MC profibrotic signaling.

### 3.5. α2M* Regulates TGFβ1 Production by HG in MCs

TGFβ1 is a well-characterized mediator of the profibrotic process in DKD and of HG-induced matrix upregulation in MCs [31,32]. Since Akt is known to regulate its synthesis in response to HG [33], inhibition of α2M* is likely to inhibit TGFβ1 production. Confirmation of this is shown in Figure 7A, in which HG-induced secretion of TGFβ1 into the medium, assessed by ELISA, was blocked by α2M* neutralization with Fα2M.

### 3.6. Urinary α2M* Is Associated with Urinary TGFβ1 in Individuals with Overt DKD

We previously showed that urinary TGFβ1 contributed to the prediction of kidney disease progression in a cohort of patients with established DKD [25]. We thus analyzed the urine of a subset of these patients to address the association between α2M* and TGFβ1. Previously, an ELISA specific to α2M* was validated for use with serum and plasma samples [22], but it has not been used for urine samples. Thus, as described in Methods, we first validated its use for human urine samples spiked with various concentrations of α2M* to confirm recovery. Figure 2B shows that the recovered α2M* from urine samples closely corresponded to the standard concentrations of α2M* used for this assay, confirming the ability of the ELISA to quantify α2M* from human patient urine samples. 

In a subset of individuals with low and high proteinuria (described in Methods), urinary α2M* was strongly associated with total urinary protein (Spearman’s Rho 0.76, *p* = 0.03, *n* = 8). To attenuate the influence of proteinuria on both α2M* and TGFβ1, we analyzed a subset of 18 patients with median proteinuria < 2 g/g of creatinine during follow-up. Table 1 shows the patient clinical characteristics. α2M* was determined in 56 samples. As shown in Figure 7C, the median value of urinary α2M* for each individual was associated with the median value of urinary TGFβ1. The association between α2M* and proteinuria was not significant in this low-proteinuria subgroup (*p* = 0.16, Pearson correlation). Together, these data identify an important role for α2M* in mediating HG-induced TGFβ1 upregulation in vitro and suggest the clinical relevance of this finding. Further clinical studies will more clearly define this relationship. 

## 4. Discussion

We recently demonstrated that csGRP78 is increased in diabetic kidneys and showed its importance in mediating MC HG-induced profibrotic signaling [7]. However, the mechanism by which csGRP78 is activated in this setting was unknown. Our data now identify α2M*, a known ligand for csGRP78, as a critical mediator of this signaling in MCs. Not only is α2M expression increased by HG in MCs and in diabetic mouse and human kidneys in both glomeruli and tubules, but more importantly, its activation enables its function as a signaling ligand for csGRP78. Localization of α2M and α2M* expression to mesangium was confirmed using ISH and IF, respectively. Similar to our previous findings with csGRP78 inhibition, α2M knockdown or α2M* neutralization inhibits profibrotic Akt activation and downstream matrix and profibrotic cytokine production. These data support an important role for α2M* in the pathogenesis of diabetic glomerulosclerosis, an early hallmark of DKD, and provide a rationale for its further evaluation as a potential therapeutic target. 

Several studies have suggested a role for α2M in diabetes and more recently in DKD. Increased α2M levels in saliva of diabetic patients corresponded with higher blood glucose levels, potentially associating α2M with glycemic control [19]. Increased α2M serum levels were also found in diabetic patients [18], and these correlated with microalbuminuria, a clinical feature associated with DKD progression [34,35]. The increased presence of α2M was further identified in diabetic kidneys by IHC, postulated to be due to non-specific leakage and trapping of plasma proteins [17]. A recent study, however, showed elevation of α2M transcript levels in diabetic glomeruli, supporting local regulation and synthesis [20]. Our studies clearly demonstrate increased local production of α2M. However, the pathogenic role of α2M in DKD was previously undefined. We now show that α2M is activated only in diabetic kidneys, where its activation is likely due to the presence of various α2M-binding proteinases in the hyperglycemic environment [36], but this needs to be further evaluated. Importantly, in this activated form, α2M* can bind to and signal through csGRP78 to induce profibrotic responses. Our data thus reveal a novel role for α2M* in the pathogenesis of DKD, which can be potentially exploited for treatment purposes.

An association of α2M with non-diabetic renal fibrosis has also been observed in several models. In age-associated renal fibrosis, α2M was increased in both glomeruli and the tubulointerstitium. It was postulated that this increase inhibited the matrix-degrading ability of MMP2, thus contributing to the accumulation of collagens in the aging kidney. With caloric restriction, α2M expression was attenuated in parallel with a decrease in fibrosis and an increase in MMP2 activity [37]. α2M was also increased in two models of renal fibrosis (puromycin aminonucleoside nephrosis and 5/6 nephrectomy), particularly in glomeruli, with expression noted to increase as fibrosis progressed [38]. Recently, increased renal α2M transcript was associated with human focal glomerulosclerosis (FSGS) disease progression and poor renal prognosis. From single-cell RNAseq data, α2M transcript was identified in both endothelial and mesangial cells [20]. Interestingly, in these studies and consistent with our data, α2M was absent in normal kidneys. We thus further investigated whether the activated α2M* was increased in our non-diabetic model of chronic kidney disease, 5/6 nephrectomy in *CD1* mice, by IHC (ethics approval number 16-07-27) [39]. Here we also observed a significant increase in α2M*, suggesting a potentially broader profibrotic role for α2M* in non-diabetic chronic kidney disease (Appendix A and separate quantification for glomeruli and tubules in Appendix A). Future studies will determine whether the cell surface presence of its receptor GRP78 is also increased in non-diabetic chronic kidney disease and whether inhibition of this pathway protects against fibrosis. 

Studies have suggested a potential for urinary α2M (the inactive form) to serve as a biomarker for DKD. One study showed a correlation between urinary α2M levels and the degree of microalbuminuria in individuals with type 2 diabetes [35]. Yang et al. identified a urine proteome specific for proliferative diabetic retinopathy, also in type 2 diabetic patients. While α2M was one of the top two proteins significantly increased in those with DKD, the prospective arm of the study focused only on the ability of haptoglobin, the second of these proteins, to predict the progression of DKD [40]. It is thus possible that urinary levels of α2M, or α2M* as suggested by our study, are potential biomarkers that may identify those with DKD and/or those at higher risk of progression. This will require further study.

α2M may promote fibrosis in two main ways. First, locally increased α2M may inhibit matrix-degrading metalloproteinases [41], thus indirectly contributing to matrix accumulation. Second, we showed that α2M* signaling through csGRP78 promotes profibrotic Akt signaling [7]. We also evaluated the contribution of LRP1, the second identified α2M* receptor [42]. Functioning as a scavenger receptor, LRP1 clears a variety of ligands through endocytosis, although it was also shown to be involved in the activation of various signaling pathways, including Akt [42]. Furthermore, LRP1 was implicated in the pathogenesis of several human diseases including Alzheimer’s disease, breast cancer, and prostate cancer [42], although it has not been directly studied in diabetes or DKD. Our data using LRP1 knockdown, however, do not support a significant role for LRP1 in MC HG-induced profibrotic responses.

The mechanism by which α2M transcript is increased by HG has yet to be elucidated. The cytokines IL-6 and IL-11 were shown to increase α2M promoter activity through members of the STAT transcription factor family [43,44]. The synergy between STAT and AP-1 was additionally important for α2M promoter regulation by IL-6 [43]. Interestingly, the activity of both STATs and AP-1 is known to be induced by HG, with the targeting of STAT activation through the use of JAK inhibitors currently being evaluated for the treatment of DKD [45,46]. The role of STAT and AP-1 in the regulation of α2M promoter activity in response to HG and in DKD, along with the potential role of other transcription factors, will be evaluated in further studies. 

Our biotinylation results suggest that locally increased α2M* may facilitate the presentation of GRP78 on the cell surface in addition to its role as a ligand, as has also been shown in prostate cancer cells [47]. This suggests that α2M* may participate in a positive feedback loop, leading to an augmentation of csGRP78 signaling and thus potentiation of profibrotic signaling. Interestingly, we observed that forced cell surface GRP78 expression was sufficient to increase Akt activation and upregulation of ECM proteins/CTGF and was augmented by HG. This was blocked by α2M* inhibition, supporting a requirement for α2M* in both basal and HG-induced signaling through csGRP78. 

Taken together, we have shown that α2M/α2M* is increased in MCs by HG and in diabetic glomeruli and that α2M* regulates MC profibrotic responses through its interaction with csGRP78 (Figure 8). These data suggest that blocking α2M*/csGRP78 interaction may be a novel therapeutic option for DKD. Importantly, our data also support the potential therapeutic use of a peptide that specifically blocks α2M*/csGRP78 interaction. Indeed, peptide therapy is well accepted in clinical use, as for example with the widely used glucagon-like peptide 1 (GLP-1) in type 2 diabetes for lowering blood glucose levels [48]. Further studies will evaluate the efficacy of a peptide targeting the α2M*/csGRP78 interaction in attenuating DKD in vivo.

## Figures and Tables

**Figure 1 biomedicines-09-01112-f001:**
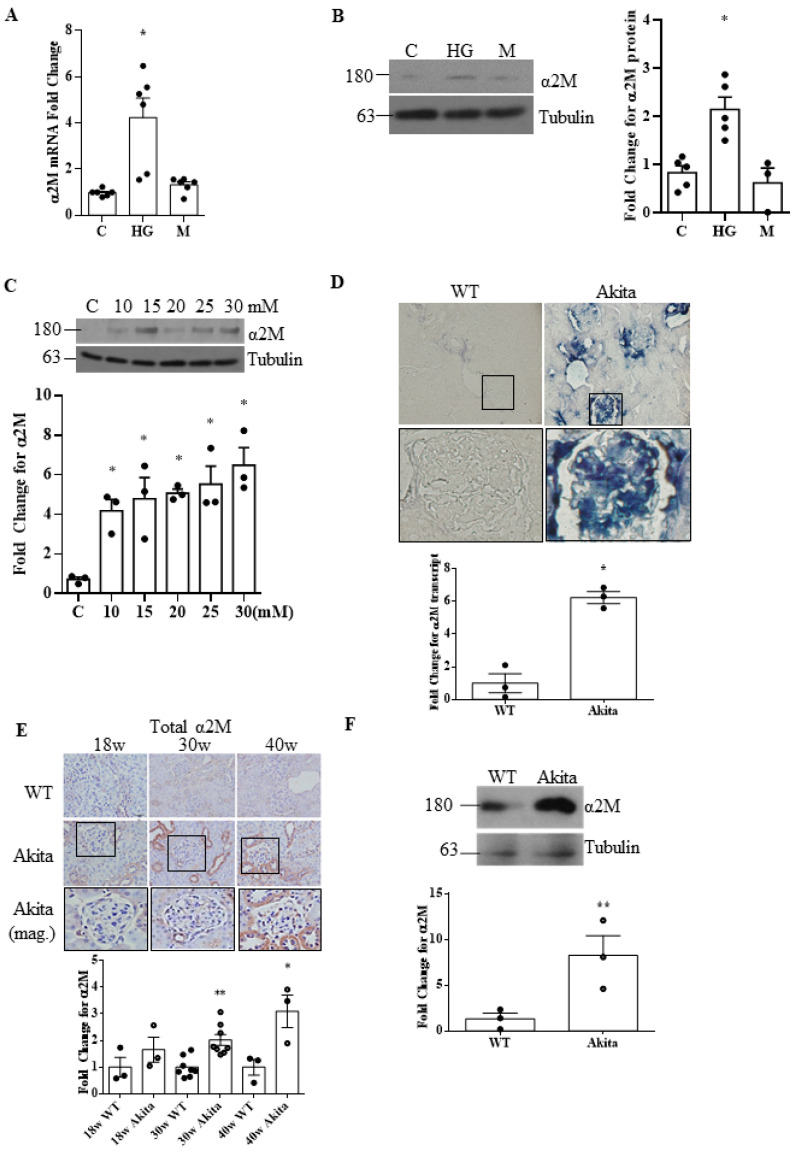
α2M is increased by HG in mesangial cells and in diabetic kidneys. HG increased α2M mRNA (24 h) (**A**) and protein (48 h) (**B**) expression in MCs. No effect was seen with the osmotic control mannitol (M) (**A**: *n* = 6; **B**: *n* = 5) (* *p* < 0.01 vs. others). (**C**) α2M expression increased dose-dependently with HG (24 h) concentrations from 10 to 30 mM (*n* = 3, * *p* < 0.01 vs. con). (**D**) α2M transcript expression, assessed by ISH, was significantly higher in type 1 diabetic *Akita* kidney sections compared to wild-type mice at 40 weeks of age (40× magnification, *n* = 3) (* *p* < 0.01 vs. control). (**E**) α2M protein expression, as assessed by IHC, was significantly higher in kidney sections of 30- and 40-week-old type 1 diabetic *Akita* mice compared to wild-type mice (40× magnification, *n* = 10). (**F**) This was also seen by immunoblotting of kidney lysate from 40-week-old *Akita* compared to wild-type mice (*n* = 3) (* *p* < 0.01 or ** *p* < 0.05 vs. respective control).

**Figure 2 biomedicines-09-01112-f002:**
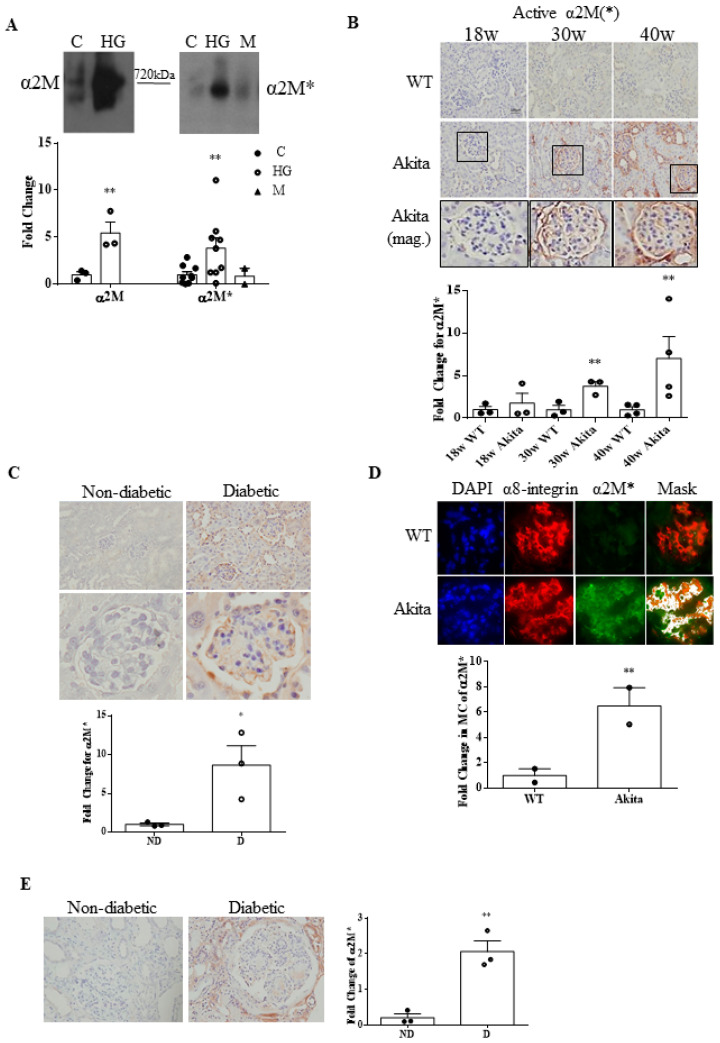
Activated α2M (α2M*) is increased by HG in mesangial cells and in diabetic kidney mesangium. (**A**) High glucose (HG, 48 h) increased media expression of α2M (left) and α2M* (right) in MCs, seen using a nondenaturing gel. Activated α2M migrates faster on a non-denaturing gel compared to its inactive form. The osmotic control mannitol (M) had no effect (*n* = 3) (** *p* < 0.05 vs. its own control). (**B**) α2M* expression was significantly higher in kidney sections from type 1 diabetic *Akita* mice compared to wild-type mice at 30 and 40 weeks of age (40× magnification, *n* = 10 each, ** *p* < 0.05 vs. respective control). (**C**) α2M* expression was also significantly higher in uninephrectomized type 1 diabetic *CD1* mice compared to their non-diabetic controls (40× magnification, *n* = 3, * *p* < 0.01 vs. control group). (**D**) α2M* colocalization with mesangial cells, identified by their marker α8-integrin, was significantly higher in kidney sections from type 1 diabetic *Akita* mice compared to wild type mice at 40 weeks of age (40× magnification, *n* = 2, 20 glomeruli per section) (** *p* < 0.05 vs. respective control). (**E**) α2M* expression in human biopsy samples from DKD patients was significantly higher compared to control kidneys (40× magnification, *n* = 4) (** *p* < 0.05 vs. control group).

**Figure 3 biomedicines-09-01112-f003:**
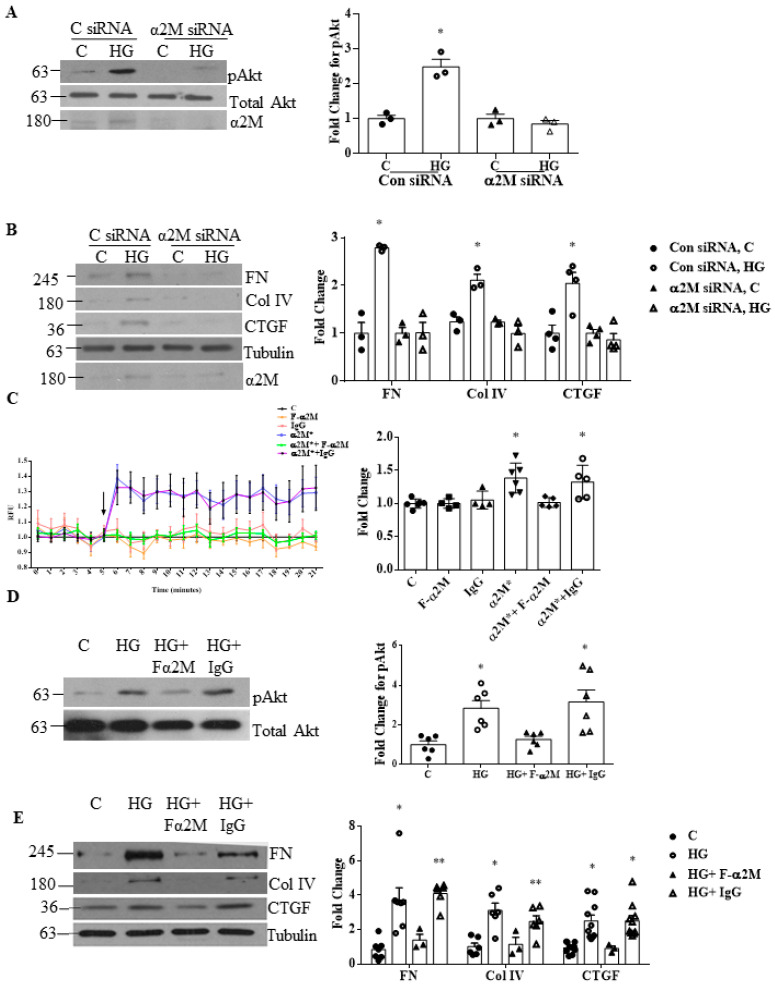
α2M knockdown or neutralization inhibits HG-induced Akt activation and downstream ECM accumulation. (**A**,**B**) Immunoblotting shows that α2M siRNA inhibited upregulation of fibronectin (FN), collagen (Col) IV, the cytokine CTGF, and Akt activation (pAkt on S473) compared to control siRNA in response to high glucose (HG, 48 h, *n* = 5) (* *p* < 0.01 vs. all others in the individual group). (**C**) Fura 2-AM calcium assay in 1-LN cells shows that the increased release of intracellular calcium stores in response to α2M* (100 pM, 15 min) was inhibited with addition of the neutralizing antibody Fα2M (2 µg/mL) but not control IgG. Groups were compared at one minute after time of treatment, which is indicated by the arrow (*n* = 6, * *p* < 0.01 or ** *p* < 0.05). (**D**,**E**) Antibody neutralization of α2M* abrogated high glucose (HG, 48 h)-induced FN, Col IV, and CTGF protein upregulation and Akt activation (pAkt on S473) compared with nonspecific IgG (*n* = 11) (* *p* < 0.01 or ** *p* < 0.05 vs. con and HG + Fα2M).

**Figure 4 biomedicines-09-01112-f004:**
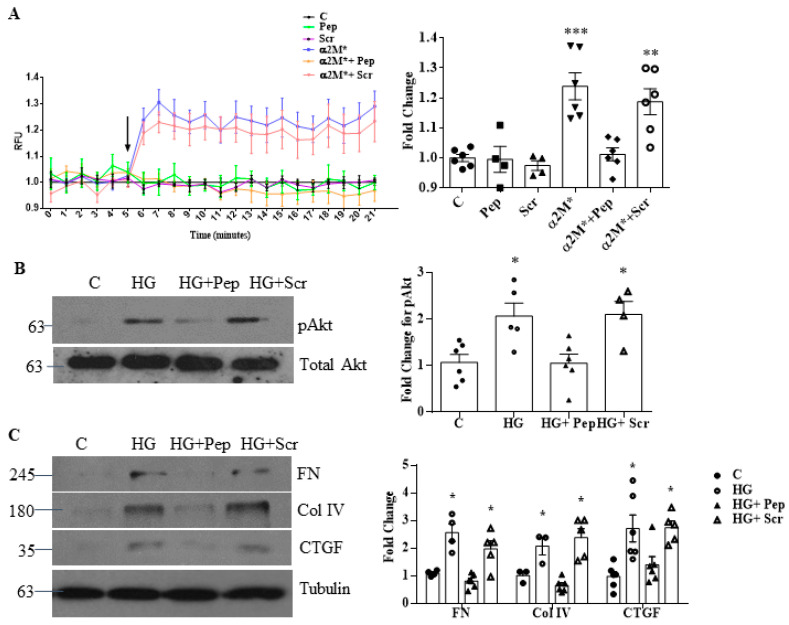
An α2M* blocking peptide inhibits HG-induced profibrotic responses. (**A**) The peptide (Pep) blocking csGRP78/α2M* interaction, but not a control scrambled (Scr) peptide (both 100 nM), inhibited release of intracellular calcium stores in response to α2M* (100 pM, 15 min) in 1-LN cells (*n* = 6, * *p* < 0.01, ** *p* < 0.05 or *** *p* < 0.001). (**B**,**C**) MCs were treated with high glucose (HG, 48 h) with or without peptides as in (**A**). The inhibitory peptide, but not the scrambled peptide, prevented HG-induced Akt activation (**B**) and upregulation of matrix proteins fibronectin (FN) and collagen (Col) IV and of the profibrotic cytokine CTGF (*n* = 5, * *p* < 0.01 vs. control and HG with peptide).

**Figure 5 biomedicines-09-01112-f005:**
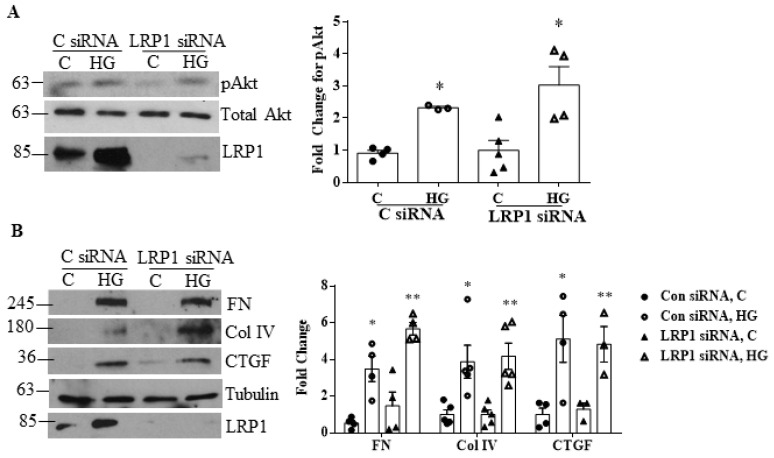
LRP1 knockdown did not affect Akt activation or ECM production. (**A**,**B**) Knockdown of LRP1 with siRNA did not attenuate Akt activation (pAkt on S473) or production of fibronectin (FN), Collagen (Col) IV, or CTGF compared to control siRNA in response to high glucose (HG, 48 h). Successful LRP1 knockdown is shown (*n* = 5, * *p* < 0.01 or ** *p* < 0.05 vs. all others in individual group).

**Figure 6 biomedicines-09-01112-f006:**
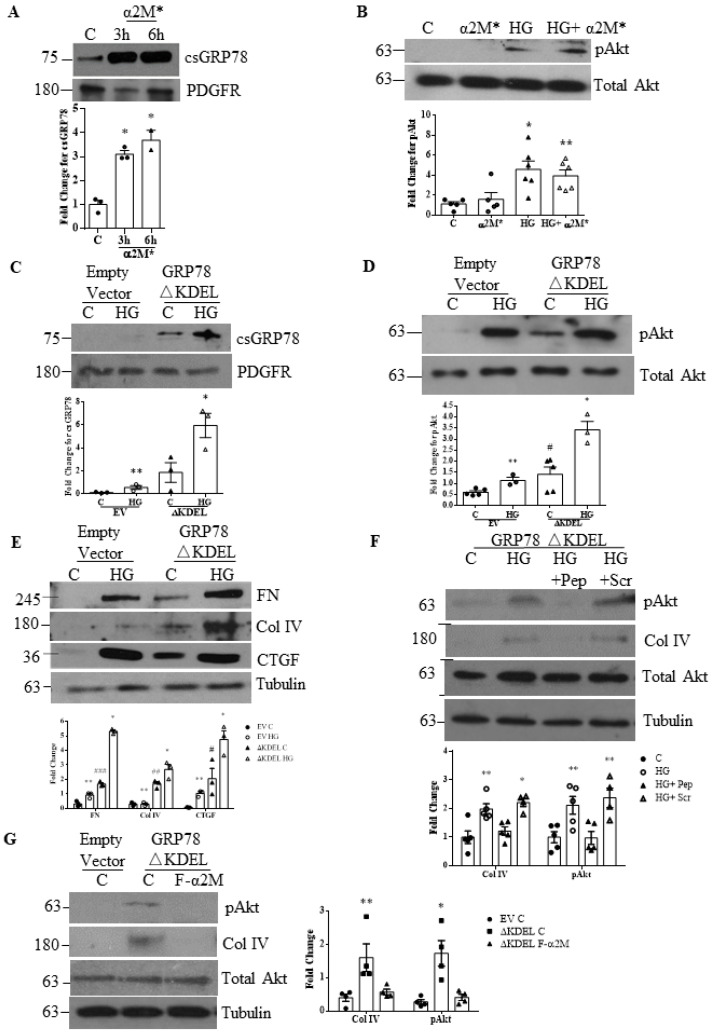
Increased matrix synthesis with csGRP78 overexpression requires α2M*. (**A**) α2M* (100 pM) induced cell surface expression of GRP78 at 3 and 6 h in MCs (*n* = 3, * *p* < 0.01) (**B**) HG-induced Akt activation (pAkt on S473) was not augmented by addition of α2M* (100 pM, 24 h) (*n* = 6, ** *p* < 0.05 or * *p* < 0.01 vs. control). (**C**) Increased translocation of GRP78 to the cell surface was induced by overexpression of GRP78 lacking KDEL (GRP78△KDEL) in MCs. (**D**) This increased both basal and high glucose (HG, 48 h)-induced Akt activation (pAkt on S473). (**E**) Similar findings were seen with HG (48 h)-induced fibronectin (FN), collagen (Col) IV, and CTGF upregulation (for **C**,**D**,**E**: *n* = 3 except for D: *n* = 5, ** *p* < 0.05 vs. empty vector control (significant through t test) or * *p* < 0.01 vs. △KDEL vector control, # *p* < 0.05 or ## *p* < 0.01 or ### *p* < 0.01 vs. empty vector control). (**F**) The augmented responses to HG seen with GRP78 △KDEL overexpression, including Akt activation (pAkt on S473) and collagen (Col) IV upregulation, were prevented by the α2M*/csGRP78 inhibitory peptide, but not the scrambled control peptide (*n* = 5, ** *p* < 0.05 or * *p* < 0.01). (**G**) Similar inhibition was seen with α2M* neutralization using Fα2M (2 µg/mL) (*n* = 4, ** *p* < 0.05 or * *p* < 0.01).

**Figure 7 biomedicines-09-01112-f007:**
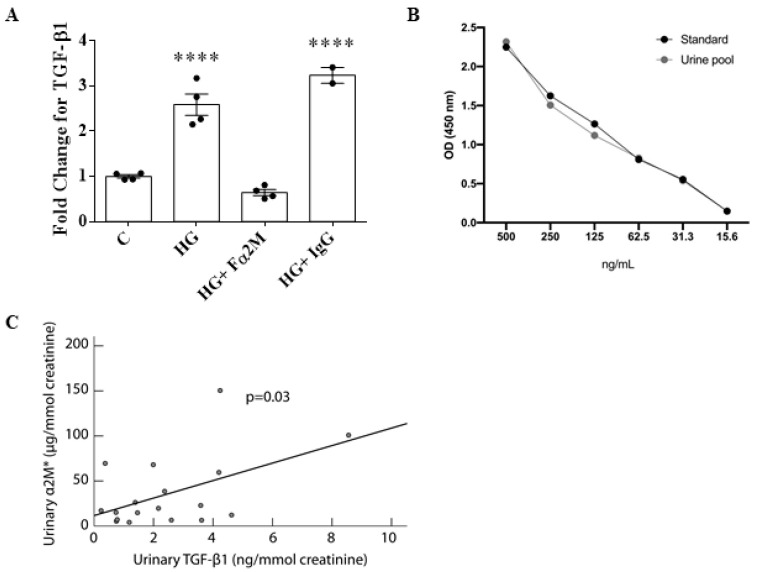
TGFβ1 production is dependent on α2M* in MCs, with urinary TGFβ1/α2M* association in patients with established DKD. (**A**) Neutralization of α2M* with Fα2M (2 µg/mL) prevented high glucose (HG, 48 h)-induced secretion of TGFβ1 into the medium, as assessed by ELISA (*n* = 4, **** *p* < 0.0001). Control IgG had no effect. (**B**) Correlation of spiked urine to standard concentrations of α2M* in human urine samples confirmed the assay’s ability to quantify α2M* in urine samples. (**C**) There was an association between urinary α2M* and TGFβ1 in patients with established DKD (*p* = 0.03, Pearson correlation).

**Figure 8 biomedicines-09-01112-f008:**
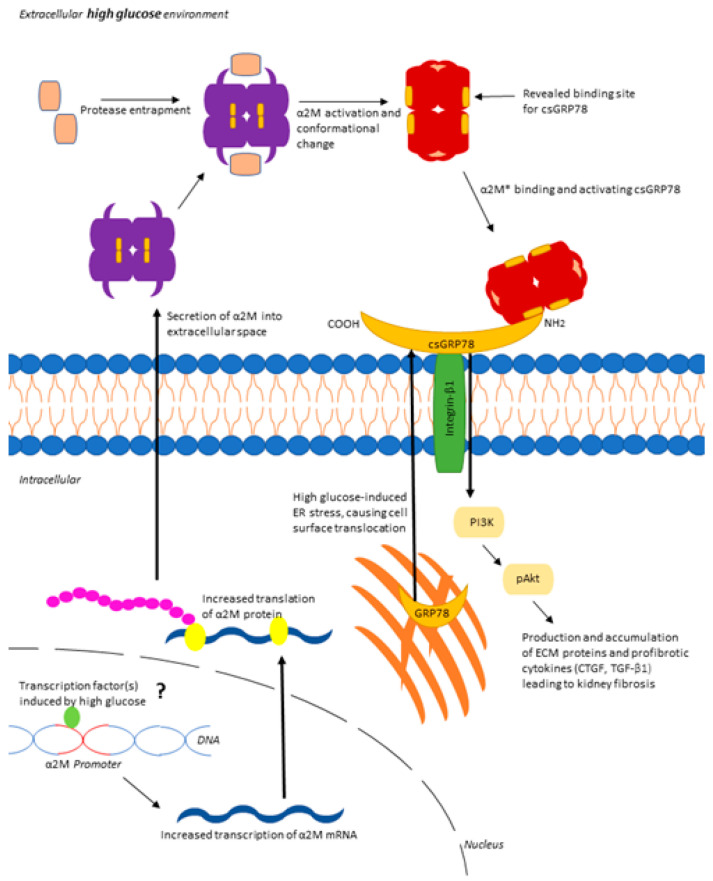
Proposed pathway for HG-induced α2M*/csGRP78 mediated profibrotic signaling in MCs. HG leads to increased synthesis (likely by HG-responsive transcription factors, but this has yet to be elucidated as indicated by “?” in the figure) and activation of α2M as well as translocation to the cell surface of its receptor GRP78. Interaction between α2M*/csGRP78 leads to activation of Akt and downstream synthesis of profibrotic cytokines and extracellular matrix proteins.

**Table 1 biomedicines-09-01112-t001:** Patient characteristics.

**Number**	18
**Male, *n* (%)**	16 (89)
**Age (years)**	70 ± 8
**Baseline eGFR (mL/min/1.73 m²)**	25 ± 10
**eGFR decline rate (mL/min/1.73 m²/year)**	−2 ± 2
**Follow-up period (year)**	2.3 ± 0.6
**Protein/creatinine ratio (g/g)**	1.3 ± 0.5
**SBP (mmHg)**	142 ± 14
**DBP (mmHg)**	66 ± 8
**RASB use, *n* (%)**	17 (94)

SBP, systolic blood pressure; DBP, diastolic blood pressure; RASB, renin–angiotensin system blocker. Normally distributed values are presented as mean ± standard deviation.

## Data Availability

The data obtained and presented in this article are available from the corresponding author upon reasonable request.

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
