# Peer review of "Activated Alpha 2-Macroglobulin Is a Novel Mediator of Mesangial Cell Profibrotic Signaling in Diabetic Kidney Disease"

_biomedicines, 2021, doi:10.3390/biomedicines9091112_

Round 1

Reviewer 1 Report

In this manuscript, authors demonstrated that activated alpha-2 macroglobulin was upregulated in mesangial cells and directly promoted the expression of extracellular matrix proteins such as type IV collage in diabetic nephropathy, probably through its receptor cell surface 78kDa glucose regulated protein (csGRP78) but not low density lipoprotein receptor-like protein (LRP). This study appears to be well designed, and the subject is interesting for many readers. However, there are some concerns mainly in data presentation. The reviewer’s comments are described as follows.

Material and Methods:

1. Authors should describe the methods of the establishment of mouse primary mesangial cells. The source of the male C57BL/6J mice should be also described.

2. Authors should describe the name of animal ethical committee that approved these animal experiments as well as the approval number that assigned by the committee. In line 110, approval number should be added. Furthermore, regarding the cohort study that verified the association between urinary activated alpha-2 macroglobulin and TGF-β, ethical committee and approval number should be described.

Results:

1. In Figure 1D-F, data showed that activated alpha-2 macroglobulin was increased not only in glomeruli but also in renal tubular cells. Since authors suggested increased expression in glomerular mesangial cells, they should separately analyze the positive areas in glomeruli and renal tubules in both in situ hybridization and immunohistochemistry. The same is true for Figure 2B, C, and E.

2. In Figure 2A, the expression of a2M and a2M* should be normalized by loading control such as tubulin.

3. In Figure 2D, these immunofluorescent images appear to be unclear and are thus unaccepted. The image resolution should be improved using laser confocal microscope.

4. In Figure 6D and E, there are obvious discrepancies between blot band images and graphs, especially in Empty vector + HG group. Appropriate band images mut be presented.

5. In Figure 6F and G, the expression of type IV collagen should be normalized by loading control such as tubulin.

6. In Figure 7, authors should explain whether activated alpha-2 macroglobulin was stable in urine enough to be accurately quantified.

Reviewer 2 Report

I am impressed with the quality of this experimental paper with some clinical implications. Very important work that expands our knowledge and undersanding of glomerular injury in DKD. Concerning the clinical part of the study authors speculate on the possible role of ∝2M-targeting molecules as a therapy of DKD. Can they comment on the role of urinary ∝2M as a biomarker to predict the rate of CKD/DKD progression? Are there any data to conclude that ∝2M - GRP78 interaction may also play a role in other glomerular diseases, other that DKD?

Round 2

Reviewer 1 Report

Authors have successfully addressed the reviewer's concerns. The reviewer thinks if authors could present higher resolution images in Figure 2D, the paper would become excellent.